# Delphi Consensus in Otolaryngology: A Systematic Review of Reliability and Reporting Completeness

**DOI:** 10.3390/jpm15120567

**Published:** 2025-11-24

**Authors:** Anastasia Urbanelli, Giorgia Pugliese, Elisa Bolis, Matilde Coccapani, Martina Gemma Corti, Barbara D’Angelo, Anna Lancieri, Laura Maggi, Antonino Maniaci, Jerome R. Lechien, Alberto Maria Saibene

**Affiliations:** 1Division of Otorhinolaryngology, Department of Surgical Sciences, Università degli Studi di Torino, 10124 Turin, Italy; anastasia.urbanelli@unito.it; 2Otolaryngology Unit, Santi Paolo E Carlo Hospital, Department of Health Sciences, Università Degli Studi Di Milano, 20142 Milan, Italyelisa.bolis@unimi.it (E.B.);; 3Faculty of Medicine and Surgery, University of Enna “Kore”, 94100 Enna, Italy; 4Department of Surgery, UMONS Research Institute for Health Sciences and Technology, University of Mons (UMons), 7000 Mons, Belgium; jerome.lechien@umons.ac.be

**Keywords:** guidelines, consensus, Delphi, methodology, bias, quality evaluation, otolaryngology

## Abstract

**Background:** The Delphi method is increasingly used in otolaryngology to develop consensus in subjects lacking robust evidence. In these contexts, consensus documents play a dual role: they provide structured expert guidance while determining shared principles adaptable to the individual patient. Nevertheless, the methodological rigor and reporting consistency of Delphi studies remain variable, raising concerns about transparency, reproducibility, and potential bias. **Methods:** A systematic review was conducted according to PRISMA guidelines. MEDLINE, Embase, and Web of Science were searched for Delphi-based consensus studies in otolaryngology. Fully published studies in English, Italian, German, French, or Spanish were included. Each article was assessed using established methodological frameworks for consensus development, bias domains described in the methodological literature, and the DELPHISTAR reporting checklist. Reliability and completeness scores were calculated to enable comparisons. **Results:** Out of 3168 unique records, 86 studies were included. Most defined their purpose and consensus criteria, but transparency regarding panel selection, anonymity, feedback, and criteria for stopping the consensus process was often missing. Based on methodological bias domains, only 9 studies (10.5%) reached a “good” reliability score, while the majority were rated as “fair.” According to DELPHISTAR, just 3 studies (3.5%) showed good levels of completeness. Reporting completeness and risk of bias proved heterogeneous across subspecialties. **Conclusions:** Delphi-based studies are increasingly shaping clinical practice in otolaryngology, but persistent methodological and reporting limitations undermine their reliability. Wider adoption of standardized frameworks is essential to improve transparency, reproducibility, and clinical impact, ensuring that consensus statements support both evidence-informed practice and personalized patient care.

## 1. Introduction

The “Delphi method” (DeMet) is a structured, group-based process used to achieve consensus among a panel of experts through multiple rounds of questionnaires. After each round, participants receive anonymous feedback summarizing the group’s responses, allowing them to revise their views in subsequent rounds. This iterative process continues until a stable consensus is reached [1].

Throughout the years, the adoption of this technique has become increasingly common for forecasting, decision-making, and developing guidelines, especially in situations where evidence is limited or uncertain [2]. The DeMet is mostly used in studies involving medicine, technology, natural science, and behavioral social sciences [3].

In medicine, Delphi-based consensus statements play a particularly crucial role, as they often represent the only structured framework available to guide clinical decisions in fields where robust evidence is scarce. In these contexts, consensus is not only a pragmatic substitute for guidelines but also a foundation for delivering personalized care: it allows clinicians to rely on expert-derived principles while tailoring therapeutic strategies to the specific needs of individual patients. This dual role makes consensus documents a cornerstone of modern personalized medicine, bridging the gap between limited evidence and the demand for individualized, evidence-informed treatments.

Given the widespread interest in the DeMet in the research field, especially in topics lacking strong evidence, several reporting guidelines have been introduced to improve methodological transparency and consistency. Notable among these is the CREDES (Guidance on Conducting and REporting DElphi Studies) [4], which provides practical recommendations for both the design and reporting of Delphi studies to ensure methodological transparency and reproducibility, and ACCORD (ACcurate COnsensus Reporting Document), which focuses on standardizing how consensus-based research is documented and interpreted [5]. However, both protocols have shown significant limitations, particularly in addressing the diversity of Delphi applications and methodological variants. To overcome these gaps, the DELPHI-STAR (Studies in The Areas of Reporting) guidelines [6] were developed, offering the most comprehensive and interdisciplinary standard currently available for reporting Delphi studies. As a result, studies interpreting the risk of bias for DeMet studies have emerged [7], allowing for a more reproducible appropriateness and reliability evaluation.

The growing adoption of the DeMet across various disciplines (including otolaryngology) highlights the need for a rigorous and standardized methodological framework: this need is also acknowledged by the AAO-HNSF (American Academy of Otolaryngology–Head and Neck Surgery Foundation) [2].

In this context, the present study carried out a systematic review of Delphi-based publications in otolaryngology to evaluate how rigorously the DeMet has been applied in the field and to identify examples of best practices as well as recurring methodological limitations. The primary aim of this systematic review is to promote more rigorous methodological standards that enhance the overall impact, clinical relevance, and scientific validity of Delphi research in this field.

## 2. Materials and Methods

This review can be found in The Open Science Framework with DOI (https://doi.org/10.17605/OSF.IO/8CXS7). Following protocol registration in the Open Science Framework database [8], we conducted a systematic review between 17 September 2024, and 5 May 2025, in accordance with the PRISMA reporting guidelines [9]. The PRISMA checklist is provided in Appendix A. Systematic electronic searches were performed in English, Italian, German, French, and Spanish to identify any Delphi-method consensus published on topics of otolaryngological interest.

On 17 September 2024, we conducted a primary search on the MEDLINE, Embase, and Web of Science databases. The search terms used were as follows: delphi AND (otolaryngolog* OR otorhinolaryngolog* OR otolog* OR rhinolog* OR laryngolog* OR audiolog* OR phoniatric* OR nose* OR nasal* OR paranasal* OR sinus* OR mouth* OR pharyn* OR laryngopharyn* OR nasopharyn* OR oropharyn* OR rhinopharyn* OR hypopharyn* OR throat* OR larynx OR laryngeal OR “vocal cord*” OR “vocal fold*” OR ear OR ears OR neck* OR salivary OR adenoid* OR tonsil* OR tongue* OR speech OR hearing OR audition* OR deglutition* OR swallow* OR respiration* OR breathing OR gustation* OR taste OR smell* OR olfaction* OR sinusitis OR rhinosinusit* OR tonsilliti* OR otiti* OR rhinitis* OR ototoxic* OR sleep* OR apnea* OR OSA OR Epistaxis OR rhinorrhea OR rhinorrea OR deafness OR hypoacus* OR hoarse* OR dysphagia OR xerostomia* OR hyposalivation* OR hyposialia).

Additionally, we planned to manually examine the references for selected publications to identify any further relevant reports not retrieved by the initial database search.

Only fully published Delphi-method consensus were considered for inclusion.

Exclusion criteria were as follows:-Non-human studies;-Non-otolaryngology-related Delphi topic;-Papers published in languages other than English, Italian, German, French, or Spanish;-Conference abstracts;-Protocol-only papers;-Studies using mixed consensus techniques or using the DeMet for purposes other than obtaining a clinical consensus (e.g., using Delphi for prioritizing items in a list);-Studies published before the Rosenfeld development manual appeared in the literature [2];-Studies lacking an otolaryngologist in the development group or with less than 50% otolaryngologists among panelists.

Abstract and full-text reviews were conducted in duplicate by different authors. During the abstract review stage, all studies deemed eligible by at least one reviewer were included. Disagreements during the full-text review stage were resolved through consensus among the reviewers.

The PICOTS framework did not apply to this study, as its research methodology was not focused on patient populations.

Data extraction and study rating were performed in duplicate by two authors, with discrepancies resolved through consensus.

Based on the frameworks developed by Rosenfeld et al. [2], Nasa et al. [7], and the DELPHISTAR initiative [6], each study was evaluated to assess its alignment with established standards for consensus-building research, to obtain a synthesized overview of the quality of the articles examined.

For each article included, the following outcomes were recorded:-The related otolaryngology subspecialty (if applicable);-Key elements such as planning, development, and structure according to the Rosenfeld development manual (defining scope, development group, appropriate literature review, modified DeMet implementation, panel inclusion criteria, number of drafting and revision rounds, number of statements, and final results) for a total of 7 items;-Potential bias elements according to Nasa et al. [7]. (identification of problem area, selection of panel members, anonymity of panelists, controlled feedback, iterative rounds, consensus criteria analysis of consensus, closing criteria, i.e., the criteria which define when the consensus process should be stopped without further rounds, stability) for a total of 9 items;-Reporting completeness according to the Delphistar protocol (38 items).

The quality assessment, which is usually a key component of a systematic review, was essentially the core component of the data extraction process and represented the primary objective of the article itself, which was specifically designed to evaluate both the reliability and completeness of reporting. Therefore, the quality appraisal is inherently embedded within those processes.

Due to the predominantly qualitative nature of the collected data, no initial or subsequent meta-analysis was planned or conducted.

In order to allow easier comparisons of the results, after evaluating each article for reliability and potential bias according to the Nasa et al. [7]. work and reporting completeness according to the Delphistar protocol, we devised and assigned a specific score.

Regarding the Nasa et al. [7]. protocol, we reported for each article a reliability score given as the percentage of the items not at risk of bias out of the nine proposed. In case an item could not be applied to a specific article (as it was often the case for the “stability” item), such an item was ignored for the overall score.

Based on the score achieved, the articles have been subsequently classified for reliability and potential bias as follows: poor: less than 35% of items not at risk of bias; fair: between 35−70% of items not at risk of bias; good: >70% of items not at risk of bias.

Following an analogous method, we also scored each article for the completeness achieved according to the Delphistar protocol and expressed it as a percentage based on the 38 proposed items. In case an item could not be applied to a particular article, such as “Information about the sources of funding” and “Justification of the Delphi variation and modifications”, those items were excluded.

Based on the score achieved, the articles’ completeness was subsequently classified as follows: very poor: less than 25% of items reported; poor: 25−50% of items reported; fair: 50−75% of items reported; good >70% of items reported.

## 3. Results

Among the 3,168 unique research items initially identified, 271 published reports were selected for full-text evaluation. No further report was identified for full-text evaluation after reference checking. Ultimately, a total of 86 articles were retained for analysis (see Figure 1). The list of DeMet consensus included in this systematic review is provided in Appendix A.

The literature was categorized according to the otolaryngology field covered: skull base, otology, laryngology, neuromonitoring, nose and paranasal sinuses, head and neck oncology, sleep and apnea, pediatric otolaryngology, thyroid, and “other”. Items retrieved for each otolaryngology topic are detailed in Table 1.

### 3.1. Application of Rosenfeld’s Methodology

All the 86 analyzed studies predetermined the purpose of the consensus according to Rosenfeld’s development manual; 65 performed and documented a literature review before the consensus, while 2 explicitly did not, and 19 did not declare it. Regarding the implementation method, 33 studies applied the classic modified method, 22 the Rosenfeld method, 15 the Nominal Group Technique (NGT), and 16 other methodologies.

The development group was explicitly described in 66 out of 86 articles: of these, 43 included an exclusively otolaryngological development group, and in 22, it was a mixed (otolaryngological and non-otolaryngological) group. A single included DeMet consensus had a completely non-otolaryngological development group. Regarding the panel, 70 articles specified the number of panelists, with a mean value of 28.7 members (median 25, interquartile range 17.75, min 7, max 125). A total of 53 articles reported a multispecialty panel group; in 29, it was an exclusively otolaryngological panel, while 4 articles did not define the clinical specialty of panel members.

84 out of 86 articles defined the number of drafting and revision rounds, with a mean value of 2.6 (median 2.5, IQR 1, min 1, max 5). 81 articles reported the number of initial and final statements. Initial statements were 50.28 on average (median 25, IQR 32, range 4–413). Approved statements were 30.76 on average (median 21, interquartile range 23, range 4–133).

### 3.2. Potential Bias Elements

In the evaluation conducted following Nasa et al. [7]. appropriateness criteria, 83 out of 86 articles correctly identified the problem area. The selection of panel members’ criteria was explicit in 47 out of 86 articles, whilst anonymity of the panelists was guaranteed only in 26 articles, with 21 articles being unclear. Controlled feedback was referred to in 6 articles, with 17 studies being unclear, whilst iterative rounds were recorded in 80 articles. Consensus criteria were established in 81 articles; analysis of consensus was described in 64. Closing criteria were established in only 16 articles, and stability in 11. Table 2 reports the reliability score of the included articles. Figure 2 provides a graphical synthesis of the same values.

### 3.3. Application of the DELPHISTAR Reporting Framework

#### 3.3.1. Title and Abstract

In 16 of the examined articles, the Delphi procedure was identified in the title, while 81 in the Delphi procedure appeared in the abstract. A total of 60 studies had a structured abstract, while 26 lacked such structuring.

#### 3.3.2. Context

The context group was subdivided into “formal” and “content” related criteria. Considering the formal-related criteria, 30 articles provided information about the sources of funding; 40 articles provided details about the team of authors and researchers, specifying the discipline and the institution; information about method consulting was made explicit in 11 articles, while information about the project background was provided in a total of 63 papers. Lastly, the study protocol was explicit in a total of 19 articles.

Regarding the content-related criteria, the justification of the chosen method to execute the consensus was explicit in 18 articles, while the aim of the Delphi procedure was written down in all the scientific productions.

#### 3.3.3. Method

The method criteria were divided into the following subgroups: body and integration of knowledge, Delphi variations, sample of experts, survey, Delphi rounds, feedback, and data analysis.

In the first subgroup, we checked if there was identification and elucidation of relevant expertise, spheres of experience, and perspectives, which were present in 58 papers; only 9 articles missed or deliberately not integrated handling of knowledge, expertise, and perspectives; lastly, in 17 studies, we found a proper definition of an expert. 

Delphi variations subgroup: there was an identification of the type of Delphi procedure and potential modifications in 70 papers; in 10 studies, there was a justification of the Delphi variation and modifications, while in 5, it was not applicable, and in the remaining 71, this was not performed.

Regarding the sample of experts: in 64 articles, the selection criteria for the experts were presented; in 85, there was the identification of the experts, and in 5.9, there was information about recruiting and any subsequent recruitment of experts.

For the survey, the elucidation of the content development for the questionnaire was performed in 83 publications, while the description of the questions with their content and structure was performed in 84 articles.

Regarding the number of Delphi rounds: they went from a minimum of 1 to a maximum of 5, with a mean value of 2.6 (median 3, IQR 3 − 2). However, the number of Delphi rounds was not explicit in 2 papers. Information about the aims of the individual Delphi rounds was discussed in 35 of the studies, and the disclosure and justification of the criteria for discontinuation were conducted for 15 scientific productions. 

In the feedback subgroup, we checked if there was any information about what data was reported back per round, and this was performed in 38 studies. 13 publications specified how the results of the previous Delphi round were fed back to the experts surveyed; it could be via frequencies, mean values, measures of dispersion, listing of comments, at the same time 3 papers specified whether feedback was differentiated by specific groups. Finally, 22 papers clarified how dissent and unclear results were handled.

Data analysis practices were one of the most concerning areas: only 2 studies outlined a detailed qualitative and quantitative strategy. However, a definition of consensus was provided much more frequently in 71 studies. Only three articles addressed weighting or subgroup analysis in their interpretation of the results.

#### 3.3.4. Results

The results criteria were subdivided into the Delphi process and the results.

Taking into consideration the Delphi process criteria, the illustration of the Delphi process was provided by 23 publications, while 8 papers reported information about any special aspects during the Delphi process. The number of experts per round, both invited and participating, was indicated in 54 cases.

Finally, for the results, we can consult the presentation of the results for each Delphi round and the final results for the 39 papers.

#### 3.3.5. Discussion

The discussion criteria are related to the quality of findings. Despite 66 studies highlighting the value of their findings, no one addressed the validity or potential transferability of their conclusions. Only 31 studies offered a critical reflection on the methodological limitations, and not a single publication attempted to assess inter-rater reliability or reproduce findings through split-panel comparisons. Table 3 reports analytic completeness data for all included articles. Figure 3 provides a graphical synthesis of the same values.

### 3.4. Overall Evaluation

#### 3.4.1. Reliability and Potential Bias According to Nasa et al. [7]

In terms of appropriateness and reliability, 12 articles were classified as “poor”, having a reliability score < 35%; 65 articles were classified as “fair” (score equal to 35–70%), and finally, only 9 articles were classified as “good” (score > 70%).

#### 3.4.2. Delphistar Reporting Completeness Score

In terms of reporting completeness, only 1 article was rated very poor (score < 25%); 38 articles were rated poor (score from 25 to 50%); 44 articles were rated fair (score 50–75%), and 3 articles were rated good, with a score > 70%.

## 4. Discussion

In recent decades, the DeMet has been widely applied across multiple disciplines, particularly in the medical and natural sciences, as well as in the behavioral and social sciences [6]. Owing to its strong ability to integrate expert opinion with available evidence, the DeMet is frequently adopted in contexts where evidence is limited or inconsistent, when primary studies are not feasible due to economic, ethical, or practical constraints, or when clinical and nursing settings present significant logistical challenges [6].

The fundamental distinction between clinical consensus statements (CCSs) and clinical practice guidelines (CPGs) has been thoroughly outlined by Rosenfeld [2]. While CPGs derive their strength from systematic reviews and randomized controlled trials, CCSs are primarily composed of expert opinions grounded in individual experience and knowledge. In this sense, CCSs are particularly valuable in contexts where the available evidence is insufficient to support the development of formal guidelines, yet considerable variability in clinical practice and opportunities for quality improvement persist. Within this framework, efforts to establish standardized approaches for managing cases lacking precise guidelines are gaining increasing relevance, especially in the era of personalized medicine, where clinicians must determine the most appropriate treatment for each patient while safeguarding their unique individuality.

In our review, which aimed to evaluate the rigor of Delphi-method consensuses within the framework of personalized medicine, we began with one of the pioneering approaches: Rosenfeld’s method. Although only 22 of the 86 articles included in our analysis explicitly adopted it, Rosenfeld’s framework remains the most standardized and consolidated methodology currently available for Delphi consensus. Endorsed by the American Academy of Otolaryngology–Head and Neck Surgery Foundation, it clearly defines key elements such as expert panel selection, questionnaire design, number of rounds, consensus thresholds, and reporting standards. This structured approach has been widely adopted in otolaryngology and other specialties, ensuring greater methodological rigor, transparency, and reproducibility compared with earlier, less formalized strategies. For these reasons, we selected it as the reference model to assess whether the articles adhered to specific methodological criteria.

The second step of our assessment concerned the application of the methodology proposed by Nasa et al. [7]. This framework introduces nine key evaluation domains—including problem identification, expert panel selection, anonymity, iterative rounds, consensus and closing criteria, and response stability—specifically designed to address the frequent methodological inconsistencies observed in Delphi studies. By structuring the process in this way, the approach improves transparency, reproducibility, and overall credibility, making it a valuable tool for both researchers and reviewers in the application of the DeMet to healthcare research.

In our review, only 9 out of 86 studies (10.5%) met enough reliability criteria to be evaluated as “good” [10,11,12,13,14,15,16,17,18,19]. This result highlights the widespread methodological weaknesses affecting Delphi studies in healthcare. Recurrent issues included unclear reporting of panel selection, absence of predefined consensus thresholds, arbitrary closing criteria, and insufficient evaluation of response stability. These shortcomings introduce a substantial risk of bias and compromise the transparency, reproducibility, and overall credibility of the consensus process.

Overall, the key methodological challenges we identified include the following:-Unclear or incomplete panel selection, often preventing assessment of the adequacy and balance of expertise;-Absence of predefined consensus thresholds, with criteria sometimes introduced post hoc or without explicit justification;-Arbitrary or undefined closing criteria, leading to uncertainty about when the Delphi process should conclude;-Lack of evaluation of response stability, making it difficult to verify whether a genuine consensus was achieved.

Finally, we applied the DELPHISTAR criteria to all 86 articles included in our review. This methodology allows an assessment of adherence to its 38-item checklist and highlights critical gaps that may compromise the reproducibility and reliability of Delphi studies in otorhinolaryngology. We observed that some articles showed relatively high compliance with the checklist, such as Craig et al. [10], who adhered to 26 criteria, and Tucci et al. [20], with 25 criteria. Conversely, other studies demonstrated poor adherence, meeting as few as 8 criteria [20,21].

This review has several limitations that should be acknowledged. First, although we adopted a comprehensive search strategy across multiple databases and languages, it is possible that relevant Delphi-based studies in otolaryngology were missing, particularly grey literature, conference proceedings, or studies published in less common languages. Moreover, we restricted our analysis to fully published articles, which may have introduced a publication bias, as studies with less rigorous methods might remain unpublished.

Second, our evaluation was inherently dependent on what was reported in the articles. We cannot exclude that some studies applied more rigorous methodological procedures but failed to describe them adequately in the manuscript. This limitation underscores the very issue our review highlights: incomplete reporting may underestimate the true quality of Delphi studies while simultaneously diminishing their reproducibility, comparability, and impact. Finally, although we applied three established methodological frameworks [2,6,7], these instruments themselves are evolving, and different evaluative criteria might have yielded partially different classifications.

When we state that an article fails to report key elements or displays a high risk of bias, we are not necessarily implying that the research is poor or the scientific work is inadequate. More often, these are outstanding and ambitious studies that suffer only from methodological or reporting shortcomings. Yet, such omissions inevitably narrow their scope and applicability, limiting the impact of what could otherwise be invaluable contributions to the field. Too often, brilliant efforts risk being overshadowed simply because they lack methodological clarity or fail to adhere to standardized frameworks.

For this reason, we believe it is imperative that all researchers in otolaryngology—and beyond—fully embrace established methodological and reporting schemes. Order generates order: adopting standardized frameworks such as Rosenfeld’s approach, Nasa’s bias domains, and the DELPHISTAR checklist ensures that consensus statements are not only expert-driven but also methodologically sound, transparent, and reproducible. In Delphi-based research, methodology represents not a secondary aspect but the true core of validity—more decisive than the evidence base itself, and even more foundational than expertise. Only through methodological rigor can these studies realize their full potential and continue to shape the future of personalized medicine.

## 5. Conclusions

This systematic review highlights both the growing role and the persistent methodological challenges of Delphi studies in otorhinolaryngology. Despite their increasing use as tools to guide clinical decision-making in areas where high-level evidence is lacking, the overall methodological rigor remains limited, with only a minority of studies meeting established quality criteria. The methodological frameworks proposed by Rosenfeld et al. [2] and Nasa et al. [7]., together with the recent DELPHISTAR reporting guidelines, provide valuable instruments to enhance transparency, reproducibility, and reliability. However, their partial and inconsistent adoption underscores the urgent need for more rigorous standardization. Enhancing methodological quality is essential not only to improve the reliability of Delphi-derived consensus statements but also to ensure their meaningful integration into the paradigm of personalized medicine, where individualized yet consistent care for each patient remains the ultimate goal.

## Figures and Tables

**Figure 1 jpm-15-00567-f001:**
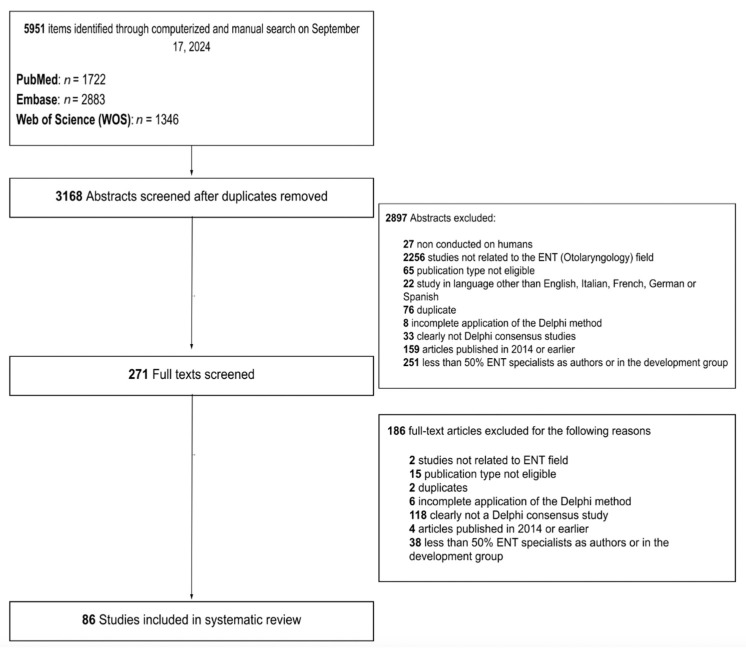
Delphi-style flowchart depicting the article selection process.

**Figure 2 jpm-15-00567-f002:**
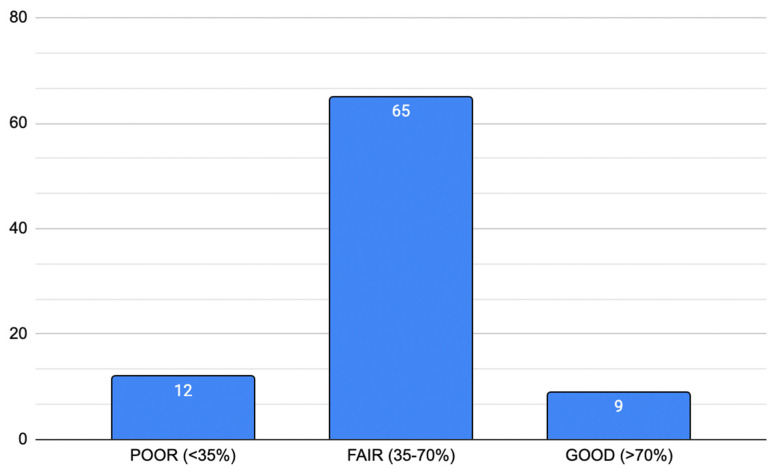
Reliability score of articles according to Nasa et al. [7]. Whenever an item could not be applied to a specific article, it was ignored for the overall score.

**Figure 3 jpm-15-00567-f003:**
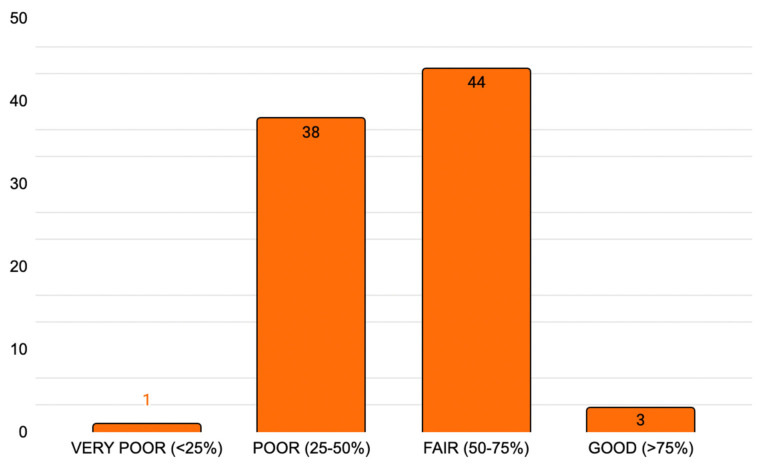
Article completeness score according to the DELPHISTAR protocol. X-axis: classification; Y-axis: total number of articles included.

**Table 1 jpm-15-00567-t001:** Topic-based categorization of the articles included in the systematic review.

Topic	No. of DeMet Consensuses Identified
Skull base	4
Ear	6
Laryngology	13
Neuromonitoring	2
Nose and paranasal sinuses	15
Head and neck oncology	13
Sleep/apnea	2
Pediatric otolaryngology	17
Thyroid	9
Other	5

The table reports the number of articles included in this systematic review according to the main topic. The section “other” includes topics addressed by a single article unclassifiable in the other sections.

**Table 2 jpm-15-00567-t002:** Article reliability.

Reliability Score According to Nasa et al. [7].								
1 Identification of problem area	2 Selection of panel members	3 Anonymity of panelists	4 Controlled feedback	5 Iterative rounds	6 Consensus criteria	7 Analysis of consensus	8 Closing criteria	9 Stability *
1/9	1/9	1/9	1/9	1/9	1/9	1/9	1/9	1/9
Total = 9/9 *								
Reliability (% of total)		Poor: score < 35%						n = 12
		Fair: score 35–70%						n = 65
		Good: score > 70%						n = 9

The table reports the 9 items of the Nasa et al. [7]. evaluation for bias and the reliability score. * If an item could not be applied to a specific article, it was ignored for the overall score.

**Table 3 jpm-15-00567-t003:** Article completeness.

DELPHISTAR Completeness Score		
1 Title and abstract	Identification as a Delphi procedure in the title; Identification as a Delphi procedure in the abstract; Structured abstract	Tot: 3/38
2 Context	Formal: information about the sources of funding *; the team of authors and/or researchers; method consulting; the project background; the study protocol	Tot: 5/38
	Content: justification of the chosen method (Delphi procedure) to answer the research question; aim of the Delphi procedure (e.g., consensus, forecasting)	Tot: 2/38
3 Method	Body and integration of knowledge: Identification and elucidation of relevant expertise, spheres of experience, and perspectives; handling of knowledge, expertise and perspectives which are missing or have been deliberately not integrated; basic definition of expert	Tot: 3/38
	Delphi variations: Identification of the type of Delphi procedure and potential modifications; justification of the Delphi variation and modifications *, including during the Delphi process, if applicable *	Tot: 2/38
	Sample of experts: Selection criteria; Identification of the experts; Information about recruitment and any subsequent recruitment of experts	Tot: 3/38
	Survey: Elucidation of the content development for the questionnaire; Description of the questionnaire	Tot: 2/38
	Delphi rounds: Number of rounds; Information about the aims of the individual Delphi rounds; Disclosure and justification of the criterion for discontinuation	Tot: 3/38
	Feedback: Information about what data was reported back per round; Information on how the results of the previous round were fed back to the experts surveyed; Information on whether feedback was differentiated by specific groups; Information about how dissent and unclear results were handled	Tot: 4/38
	Data analysis: Disclosure of the quantitative and qualitative analytical strategy; Definition and measurement of consensus; Information on group-specific analysis or weighting of experts	Tot: 3/38
4 Results	Delphi process: Illustration of the Delphi process; Information about special aspects during the Delphi process; Number of experts per round (both invited and participating)	Tot: 3/38
	Results: Presentation of the results for each Delphi round and the final results	Tot: 1/38
5 Discussion	Quality of findings: Highlighting the findings from the Delphi study; Validity of the results; Reliability of the results; Reflection on potential limitations	Tot: 4/38
Total = 38/38 *		
Completeness (% of total)	Very poor: score < 25%;	n = 1
	Poor: score 25–50%;	n = 38
	Fair: score 50–75%;	n = 44
	Good: score > 70%	n = 3

The table reports the 38 items of the DELPHISTAR protocol and the article completeness score. * If an item could not be applied to a specific article, it was ignored for the overall score.

## Data Availability

The original contributions presented in this study are included in the article/Appendix A. Further inquiries can be directed to the corresponding author.

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
