# Peer review of "Delphi Consensus in Otolaryngology: A Systematic Review of Reliability and Reporting Completeness"

_jpm, 2025, doi:10.3390/jpm15120567_

Round 1

Reviewer 1 Report

Comments and Suggestions for Authors

I thank the authors for their interesting manuscript, which I trust will be of interest to the fields of otorhinolaryngology and consensus methodology more generally. I have a number of minor suggestions to improve the manuscript.

Introduction:

- You introduce several frameworks for methodological robustness and reporting standards. Please define these in detail and present your justification for their use.

Methods:

- Consider reporting the inclusion and exclusion criteria in a table or a list of bullet points.

- Line 159: There is some text from the template remaining, please remove this.

Results:

- Line 219: Please provide definitions for "formal" and "content"-related criteria.

Discussion:

- In the abstract and conclusion, you highlight "persistent methodological challenges", but in the discussion you mostly comment on reporting. Please explicitly describe the methodological challenges you have identified.

- Line 373: There is some text from the template remaining, please remove this.

Author Response

First of all, I would like to thank the reviewer for all the insights he provided to our work. Here are the responses to all the comments.

  • You introduce several frameworks for methodological robustness and reporting standards. Please define these in detail and present your justification for their use.
    --> In the Introduction section, from Line 64 to Line 68, we provided clearer definitions and justifications for each framework mentioned (CREDES, ACCORD). As for the DeMet, its structure, rationale, and application criteria are further detailed in the Methods and Results sections, where each domain of the framework is systematically applied and discussed also in relation to our review-work itself.
  • Methods: Consider reporting the inclusion and exclusion criteria in a table or a list of bullet points.
    --> In the Methods section, we reported the inclusion criteria in a list of bullet point. Thank you for your advise.

    Line 159: There is some text from the template remaining, please remove this.
    --> Thank you very much, we deleted that.

  • Results: Line 219: Please provide definitions for "formal" and "content"-related criteria.
    --> Actually, the definitions of both "formal" and "content"-related criteria are already provided immediately after their first mention in the text. Specifically, the clarifications for "formal-related" criteria in the text are: the sources of funding; the team of authors and researchers specifying the discipline and the institution; information about method consulting; information about the project background; study protocol.

    As for the "content-related" criteria in the text: the justification of the chosen method to execute the consensus; while the aim of the Delphi procedure.

  • Discussion: In the abstract and conclusion, you highlight "persistent methodological challenges", but in the discussion you mostly comment on reporting. Please explicitly describe the methodological challenges you have identified.
    --> We thank the reviewer for this insightful comment. We added a paragraph in the Discussion section (from Line 344 to Line 352) to clarify the methodological challenges.

    - Line 373: There is some text from the template remaining, please remove this.
    --> Thank you, we removed it.

Reviewer 2 Report

Comments and Suggestions for Authors

Thank you for submitting your work.
Overall it is an adequately planned study , which aimed to evaluate the rigor of Delphi-method consensuses, published so far in the areas of interest of otorhinolaryngology. The Delphi consesnsus method is gaining popularity in the context of personalized medicine.

Few minor comments to address c:
-please delete the section" .3. Results 158
This section may be divided by subheadings. It should provide a concise and precise 159
description of the experimental results, their interpretation, as well as the experimental 160
conclusions that can be drawn. 161",

which should not appear in the final version of the manuscrript
as well as further in the article:
e.5. Conclusions 373
This section is not mandatory but can be added to the manuscript if the discussion is 374
unusually long or complex. 375"

Author Response

Thank you for submitting your work.
Overall it is an adequately planned study , which aimed to evaluate the rigor of Delphi-method consensuses, published so far in the areas of interest of otorhinolaryngology. The Delphi consesnsus method is gaining popularity in the context of personalized medicine.
--> Thank you very much for your kind and constructive comments. We really appreciate your feedback.

-please delete the section" .3. Results 158
This section may be divided by subheadings. It should provide a concise and precise 159
description of the experimental results, their interpretation, as well as the experimental 160
conclusions that can be drawn. 161", which should not appear in the final version of the manuscript
--> We have deleted it, thank you.

- as well as further in the article:
e.5. Conclusions 373
This section is not mandatory but can be added to the manuscript if the discussion is 374
unusually long or complex. 375"
--> We also deleted this wrong sentence, thank you.

Reviewer 3 Report

Comments and Suggestions for Authors

The authors describe an assessment of consensus methodology, based on the techniques described by Rosenfeld, Nasa, and DELPHISTAR. 

I like the spirit of this manuscript, which invokes rigor and consistency in the important process of consensus statement development. It is somewhat like a bibliometrics study which evaluates the current state of specific types of literature. It seems to conclude that the current work in the literature is overall fair, at least based on the figures shown.

"Results: Of 3,168 unique records, 86 studies were included. Most defined their purpose and consensus criteria, but transparency regarding panel selection, anonymity, feedback, and closing criteria was often..."
-The term "closing criteria" is unclear, at least at the point of the abstract. Please consider other terminology for the abstract and a more clear definition in the manuscript itself.

"Given a widespread interest in the DeMe in the research field, especially in topics"
-Suggest consistency in your acronym: either DeMe or DeMet 
-Also line 390

One dichotomy is that consensus statements are often developed to advise practice across large populations, which may be inherently at odds with the concept of personalized medicine, in which individuals receive more targeted care which may actually differ from person to person. While the authors allude to personalized medicine and consensus statements about personalized medicine, this point remains a question in my mind. I will defer to the journal leadership about whether the topic suits the mission of this journal. 

Author Response

I like the spirit of this manuscript, which invokes rigor and consistency in the important process of consensus statement development. It is somewhat like a bibliometrics study which evaluates the current state of specific types of literature. It seems to conclude that the current work in the literature is overall fair, at least based on the figures shown.
--> Thank you very much for your thoughtful and encouraging comments. We truly appreciate your feedback and are glad that you recognize the intent and approach of our study.

"Results: Of 3,168 unique records, 86 studies were included. Most defined their purpose and consensus criteria, but transparency regarding panel selection, anonymity, feedback, and closing criteria was often..."
-The term "closing criteria" is unclear, at least at the point of the abstract. Please consider other terminology for the abstract and a more clear definition in the manuscript itself.
--> closing/stopping???

- Thank you for pointing out this issue. We chose a different wording for the abstract and clarified better the concept of closing criteria in the methods section

  • "Given a widespread interest in the DeMe in the research field, especially in topics"
    -Suggest consistency in your acronym: either DeMe or DeMet 
    -Also line 390
    --> Thank you for the comment; we have standardized all acronyms to DeMet.Thank you for the comment; we have standardized all acronyms to "DeMet".